# 1/4 is the new 1/2 when topology is intertwined with Mottness

Peizhi Mai [1], Jinchao Zhao [1], Benjamin E. Feldman [2,3,4] & Philip W. Phillips [1] ✉

In non-interacting systems, bands from non-trivial topology emerge strictly at half-filling and exhibit either the quantum anomalous Hall or spin Hall effects. Here we show using determinantal quantum Monte Carlo and an exactly solvable strongly interacting model that these topological states now shift to quarter filling. A topological Mott insulator is the underlying cause. The peak in the spin susceptibility is consistent with a possible ferromagnetic state at $T = 0$. The onset of such magnetism would convert the quantum spin Hall to a quantum anomalous Hall effect. While such a symmetry-broken phase typically is accompanied by a gap, we find that the interaction strength must exceed a critical value for this to occur. Hence, we predict that topology can obtain in a gapless phase but only in the presence of interactions in dispersive bands. These results explain the recent quarter-filled quantum anomalous Hall effects seen in moiré systems.

Although topological insulators[1–16] represent a new class of bulk insulating materials with gapless conducting edges, their physics is completely entailed by the band theory of non-interacting electrons. The new twist is that should two atoms reside in each unit cell, the standard insulating gap that obtains at half-filling, full lower band, does not tell the whole story when spin-orbit coupling[1–3] is present. As long as time-reversal invariance is maintained, two spinful counter-propagating edge modes exist and exhibit a quantized conductance proportional to $e^2/h$, thereby giving rise to a quantum spin Hall (QSH) effect in two dimensions. Within the Kane–Mele (KM)[1,2] and Bernevig–Hughes–Zhang (BHZ)[3] models, the QSH effect obtains only at half-filling. In a general non-interacting system, this physics obtains at a filling equal to the inverse number of atoms per unit cell, $1/q$. This physics is robust to perturbations that yield only smooth deformations[16] of the Hamiltonian. Additionally, the quantum anomalous Hall (QAH) effect, that is, the existence of a quantized Hall conductance with zero net magnetic field, also requires half-filling of the Haldane model[17]. As the QAH effect breaks time-reversal symmetry while the QSH effect does not, it is difficult for them to be realized in the same material.

However, recently, both effects[18,19] have been observed in the same material in direct contrast to predictions of standard non-interacting models[1–3]. In the AB-moiré-stacked transition metal dichalcogenide (TMD) bilayer MoTe$_2$/WSe$_2$[18,19], the QSH insulator is observed at $v = 2$ with the QAH effect residing at $v = 1$. To date, this constitutes the first observation of the intertwining of these effects in the same material and hence the question of the minimal model required to explain the conflation of both is open. In terms of the 4-band KM/BHZ model, $v = 2$ and $v = 1$ correspond to half-filling and quarter-filling, respectively. Numerous theories[20–31] have been put forth in this context, and the most recent experiment[32] shows that both valleys contribute to the QAH effect and hence valley coherence rather than valley polarization is the operative mechanism. The striking deviation from the standard theory raises the question: can interactions drive either of these transitions away from half- to quarter-filling in the KM/BHZ models?

It is this question that we address here. We show quite generally that at a temperature above any ordering tendency, strong interactions shift the QSH effect to quarter filling with a decrease of the spin Chern number by a factor of two. However, the spin susceptibility exhibits a peak indicating a tendency to ferromagnetism as the temperature is lowered. Such an ordered ground state would be consistent with the Lieb–Schultz–Mattis[33,34] (LSM) theorem and recent exact

[1]Department of Physics and Institute of Condensed Matter Theory, University of Illinois at Urbana-Champaign, Urbana, IL 61801, USA. [2]Geballe Laboratory of Advanced Materials, Stanford, CA 94305, USA. [3]Department of Physics, Stanford University, Stanford, CA 94305, USA. [4]Stanford Institute for Materials and Energy Sciences, SLAC National Accelerator Laboratory, Menlo Park, CA 94025, USA. ✉e-mail: dimer@illinois.edu

diagonalization[35] on one of the models treated here. Whether or not such a ground state is gapped depends on the flatness of the band and interaction strength. In the flat-band limit, the ferromagnetic ground state is always gapped whereas for a dispersive band, the interactions must exceed a critical value for a gap to obtain. These results raise the possibility of a gapless topological semi-metallic state with non-trivial temperature corrections to the Hall conductance[36]. Generally, We argue that when the interactions dominate, the QSH must give way to a ferromagnetic QAH state at $T = 0$ at 1/4-filling. Since this is a generic conclusion on the most general models proven to undergird the QSH effect, we analyze the experiments[19,32] in this context. Our model yields a quarter-filled QAH effect which coexists with a QSH effect at half-filling as is seen experimentally, in the presence of a flat lower band and intermediate interaction.

A brief survey of interacting topological systems is in order as our key result hinges on the interplay between the two. Most studies on the KM-Hubbard[37–40] and the BHZ-Hubbard[41–44] models focused on the half-filled system and found a transition from a QSH insulator to a topologically trivial anti-ferromagnetic Mott insulator as the interaction strength $U$ increases. In addition, for models more relevant to flat-band twisted bilayer graphene, refs. 45,46 have provided a strong-coupling analysis and a density-matrix-renormalization group study[47] has found that the gapless state at half-filling in the spinless (and hence Mottless) Bisritzer–MacDonald (BM) model[48] yields a quantum anomalous Hall state in the presence of Coulomb interactions. In an extensive[49] exact diagonalization study on an 8-band BM model, $U(4)$ ferromagnets were observed always with the onset of a gap. Quantum Monte Carlo[50,51] on the spinful model reveals a series of insulating states at half-filling. In the mean-field context, models focused on layered graphene systems have addressed the origin of quantum Hall ferromagnetism in the interacting BM model[45,52,53] while others have argued that a topological Mott insulators (TMI) emerges at half-filling in the presence of on-site and nearest neighbor interactions in the tight-binding model (with only nearest-neighbor hopping) on a honeycomb lattice[54]. However, the latter proposal has not been substantiated by subsequent numerical studies[55–58] that have found half-filling to be a trivial Mott insulator when interactions are sufficiently large. Interactions also lie at the heart of fractional topological insulators[4,10,59–61] built from fractional Chern insulators[62–66] which resemble the fractional quantum Hall effect but with no net magnetic field. Such phases appear at a fractional filling in a flat-band $\Delta_0 \gg W_0$ (where $\Delta_0$ is the non-interacting topological gap and $W_0$ is the bandwidth) and require nearest-neighbor interactions. A recent study on the strongly interacting spinful Haldane model[67] demonstrates that a Chern Mott insulator originates at quarter-filling with Chern number $C = \pm 1$. This physics arises as a general consequence of an interplay between Mottness and topology.

Motivated by refs. 19,32,67, we explore the general phenomena that emerge from the interplay between Mottness and the QSH effect in the context of the KM and BHZ models. To demonstrate that the quarter-filled state is a TMI with a strongly correlated QSH effect, we numerically solve both the KM-Hubbard and BHZ-Hubbard Hamiltonians using determinantal quantum Monte Carlo (DQMC) as well as dynamical cluster approximation (DCA) and construct an analytically solvable Hamiltonian for a general interacting QSH system and obtain consistent results for sufficiently large interactions.

## Results

### Hubbard interaction

The DQMC simulation results for the generalized KM–Hofstadter–Hubbard (KM-HH) model (see "Methods") on a honeycomb lattice at $\psi = 0.81$ and $t'/t = 0.3$ are shown in Fig. 1. For this choice of parameters, the non-interacting lower band is rather flat with bandwidth $W_{0-} \approx 0.28$ and the topological gap is $\Delta_0 \approx 1.62$, the upper bandwidth is $W_{0+} \approx 4.37$, where the subscript 0 indicates non-

interacting. This mimics the flat-bands in moiré TMD experiments. The tunability of bandwidths in the KM model (unlike the bands in the BHZ model which are always dispersive $W_{0+} = W_{0-} \geq \Delta_0$) makes the KM model ideal for studying both flat-band and dispersive physics.

A key quantity that helps discern the topology in the presence of a probe magnetic field is the charge compressibility,

$$\chi = \beta\chi_c = \frac{\beta}{N}\sum_{i,j}\left[\langle n_i n_j\rangle - \langle n_i\rangle\langle n_j\rangle\right], \quad (1)$$

where the sublattice and spin summations are implied in $n_i$. Regardless of density, the inverse slope of the leading straight-line incompressible valley that extends to the zero-field limit[67] provides the Chern number. As a probe, this field does not alter our claim of a QSH phase at zero field. In the non-interacting case (Fig. 1a) at $\beta = 7$, there is a short middle vertical straight line at low fields which indicates a Chern number $C_0 = 0$ at $\langle n\rangle = 2$. This state bifurcates into two lines or equivalently two Landau levels (LLs) at higher magnetic flux. This crossing pair of zero-mode LLs is a reliable fingerprint for the QSH effects observed in experiments[11]. Note the asymmetry around $\langle n\rangle = 2$ arises entirely because the lower band is flat while the upper band is dispersive. In this regime, the lines with finite slopes all represent the standard integer quantum Hall states.

The second quantity we calculate is the spin susceptibility defined as

$$\chi_s = \sum_r S(r) - Nm_z^2 = \frac{1}{N}\sum_{i,r}\left[\langle S_i^z S_{i+r}^z\rangle - \langle S_i^z\rangle\langle S_{i+r}^z\rangle\right], \quad (2)$$

where $m_z = \sum_i\langle S_i^z\rangle/N$ is the magnetization per spin. The non-interacting spin susceptibility is related to the compressibility by $\chi_s = \chi/(4\beta)$ as shown in Fig. 1b with reverse color scale. Fig. 1c shows the magnetization. Even though the Zeeman field is absent, a non-zero Peierls flux can magnetize the system since the spin-up and -down electron bands have different Chern numbers. The non-interacting results at lower temperatures can be found in the Supplement. What we alert the reader to is the absence of any topologically non-trivial states at $\langle n\rangle = 1$.

In the presence of interactions $U = 3t$ (already strongly correlated for the lower band), the new feature and hence prediction is the emergence of a topologically non-trivial state at $\langle n\rangle = 1$. In Fig. 1d, the inverse slope of the trace extending to $\langle n\rangle = 1$ is $\pm 1$ and thus gives the Chern number. The absence of the right-moving counterpart signifies a QAH effect rather than a QSH effect. At $\langle n\rangle = 2$, the standard QSH effect remains. Consequently, we have a system in which both the QAH and QSH effects obtain simply by changing the filling. For $\langle n\rangle > 2$, the physics is weakly interacting as $U < W_{0+}$. The bright peak in the spin susceptibility in Fig. 1e indicates a possible tendency for ferromagnetism at $\langle n\rangle = 1$. This is supported by the asymmetry in the dotted lines that cross at zero field and $\langle n\rangle = 1$ in the magnetization in Fig. 1f. Such asymmetry signifies that an infinitesimal field would lead to a polarization of the spins and hence ferromagnetism.

We then further increase the interaction strength but have to raise the temperature to $\beta = 3$ due to the Fermion sign problem in DQMC (see Supplement). In the final row of Fig. 1 for the compressibility when $U = 12t$, which far exceeds $W_{0-} + W_{0+} + \Delta_0 \approx 6$, the non-interacting QSH Landau fan vanishes for $\langle n\rangle = 2$ turning into a trivial Mott insulator and most strikingly, a new LL emerges corresponding to the mirror image of the QAH state that terminates at $\langle n\rangle = 1$. The presence of both Landau components completes the high-temperature QSH features at quarter filling. The magnetization (Fig. 1i) shows a more dramatic change than does the compressibility; namely it vanishes at $\langle n\rangle = 2$ as a result of the anti-ferromagnetic Mott insulator. Further, the magnetization splits into peaks on either side of $\langle n\rangle = 1$ that continues to be asymmetrical and hence is consistent with a tendency for spontaneous

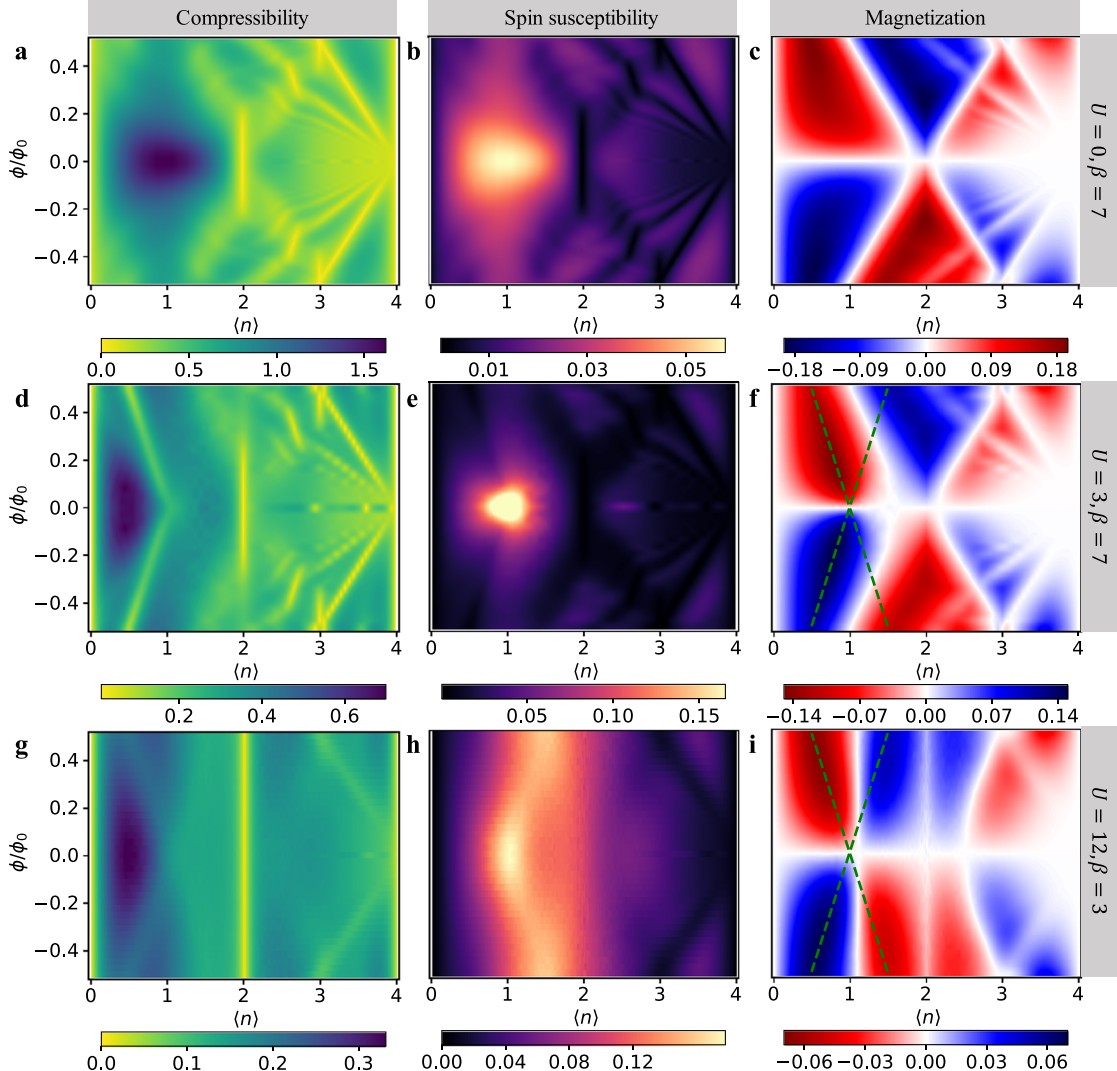

**Fig. 1 | Compressibility, spin susceptibility and magnetization of the flat-band generalized KM-HH model.** DQMC results for the flat-band generalized KM-HH model ($t' = 0.3, \psi = 0.81$) at $U = 0, \beta = 7/t$ (**a–c**), $U = 3t, \beta = 7/t$ (**d–f**) and $U = 12t, \beta = 3/t$ (**g–i**). In each row, the compressibility, spin susceptibility and magnetization are presented in order from left to right as a function of magnetic flux and electron density. The dashed green lines in panels **f** and **i** serve as a guide to the eye for the crossing pairs of Landau levels in the QSH effect.

Ising ferromagnetism despite the presence of both LLs. This physics in Fig. 1d–f is only present in the flat-band limit when $U$ is much larger than the bandwidth but comparable to the topological gap. Consequently, our theoretical work here is consistent with the sudden onset of the QAH state. Since the temperature for Fig. 1g–i is higher than the previous row, their features are softer.

To confirm the tendency for ferromagnetism, it is important to compute the temperature dependence of the spin susceptibility. Shown in Fig. 2a is the inverse spin susceptibility as the temperature is lowered with zero external magnetic flux. Displayed clearly is a possible divergence of the susceptibility ($1/\chi_s \to 0$) consistent with ordering. With extrapolation, we find that it supports a finite-temperature transition to ferromagnetism. Note that this does not violate the Mermin–Wagner theorem which forbids the spontaneous breaking of continuous symmetries at finite temperature in low-dimensional ($d \leq 2$) systems with short-range interactions. In the KM-Hubbard model with spin-orbit coupling, the system no longer has the full SU(2) symmetry but only conserves $\hat{S}^z$. Then it is the Ising symmetry that is spontaneously broken in this transition and thus allowed at a finite temperature. As this is an interaction-driven effect, we expect an enhancement of the susceptibility as $U$ increases. This is also borne out

in Fig. 2b. Together these figures justify our claim of interaction-driven ferromagnetism as the temperature is lowered. A ferromagnetic QAH state will stabilize at zero temperature even though QSH features could be present at high temperatures when $U$ is sufficiently large (Fig. 1g–i). We also observe a similar high-temperature phenomenon in the dispersive case $\psi = 0.5$ (see Supplement).

To show the generality of the 1/4-filled topological state, we consider the BHZ-Hofstadter-Hubbard (BHZ-HH) model (see "Methods") on a square lattice. Note in this model, both bands are dispersive and have the same bandwidth. Without loss of generality, we set $M/t = 1$, then $W_{0-} = W_{0+} = \Delta_0 = 2t$ ($t = 1$ as the energy scale). The non-interacting 1/2-filled system is a QSH insulator with $C_s = 2$. It is the spin Chern number that describes a QSH insulator. To measure this quantity, we use a spin-dependent time-reversal-invariant (TRI) magnetic field inspired by cold-atom experiments[68,69], namely $\phi_{\mathbf{i,j}} \to \sigma\phi_{\mathbf{i,j}}$. The compressibility measured in this way we refer to as TRI compressibility. The minus sign coupled to spin-down electrons changes the corresponding Chern number $C_\downarrow^{\text{TRI}} = -C_\downarrow$. Thus, the "TRI" Chern number measured in the TRI compressibility $C^{\text{TRI}} = C_\uparrow^{\text{TRI}} + C_\downarrow^{\text{TRI}} = C_\uparrow - C_\downarrow = C_s$ corresponds to the spin Chern number in the BHZ-HH model. This method overcomes the breakdown of the simple additivity

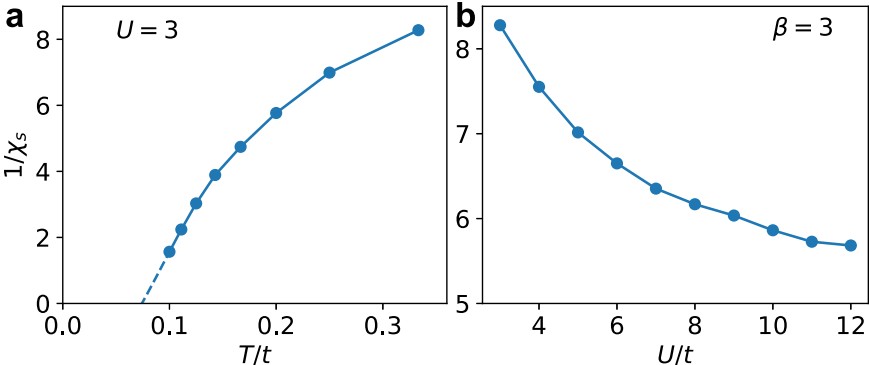

**Fig. 2 | Temperature evolution and *U*-dependence of the spin susceptibility.** Inverse spin susceptibility $1/\chi_s$ at quarter-filling ($\langle n \rangle = 1$) of the interacting flat-band generalized KM-HH model. **a** contains the temperature evolution of $1/\chi_s$ at $U/t = 3$ with extrapolation to zero. **b** shows $1/\chi_s$ as a function of interaction strength at a fixed inverse temperature $\beta = 3/t$.

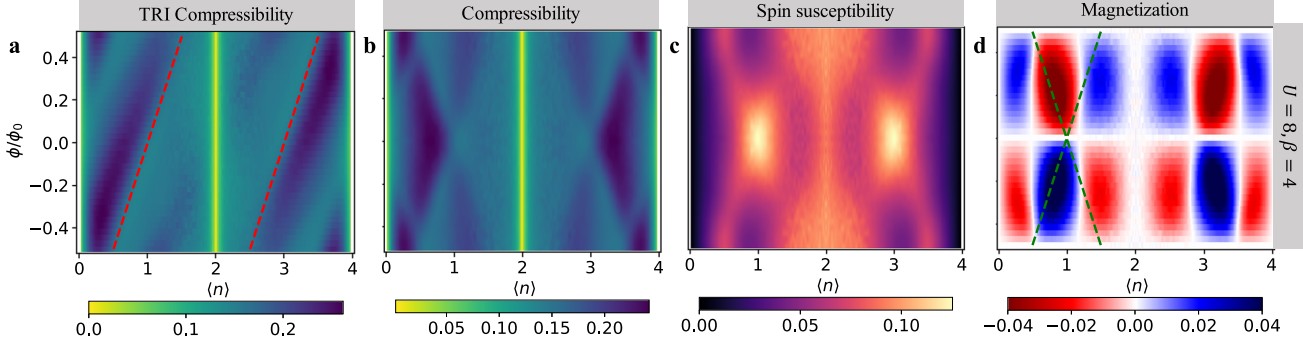

**Fig. 3 | DQMC simulations of the BHZ-HH model.** DQMC results for the TRI compressibility (**a**), compressibility (**b**), spin susceptibility (**c**), and magnetization (**d**) of the BHZ-HH models at $U/t = 8$, $\beta = 4/t$. The dashed green lines in (**d**) serves as a guide for the crossing Landau levels signaling the QSH effect.

formula for $C_s$ when the spin channels are mixed and **k** is no longer a good quantum number in the presence of interactions. Through this quantity, we can read the spin Chern number from the inverse slope of the TRI compressibility (see the Supplement for the non-interacting examples).

The simulation results for the BHZ-HH models at $U = 8t$, $\beta = 4/t$ are presented in Fig. 3. In Fig. 3a, two (red) straight lines appear from the zero-field $1/4-$ and $3/4-$filled system, whose inverse slope indicates that the corresponding zero-field $1/4-$ and $3/4-$filled BHZ-HH systems present QSH feature with $C_s = 1$ while the $1/2-$filled system becomes a topologically trivial Mott insulator with $C_s = 0$. This physics becomes much clearer by studying the standard charge compressibility in Fig. 3b which reveals identical features at $\langle n \rangle = 1$ and $\langle n \rangle = 3$ of left and right moving LLs indicative of the QSH effect. Also, the spin susceptibility exhibits a peak both at $\langle n \rangle = 1$ and $\langle n \rangle = 3$. The simultaneous appearance of compressibility minima and spin-susceptibility maxima are key features of this Mottness-driven QSH effect, in contrast to its non-interacting counterpart. The magnetization in Fig. 3d is also asymmetrical indicating a possible tendency towards ferromagnetism at $\langle n \rangle = 1$ and $\langle n \rangle = 3$. We return to this in a later section.

To corroborate our findings, we conducted a finite-size analysis (see Supplement) and confirm that the same spin Chern number survives in system sizes as large as $N_{site} = 12 \times 12$ with insignificant finite-size effects and hence our results are valid in the thermodynamic limit. We conclude then that the DQMC exhibits the QSH effect at high temperatures at 1/4-filling when $U$ is sufficiently large.

### Exactly solvable model for interacting quantum spin Hall insulators

The natural question arises: why is 1/4-filling the new topologically relevant filling and can it be understood in a simple way? The answer is

yes. For a system with 2 atoms per unit cell, there should be interaction-induced insulating states at any integer filling up to 4 charges in each unit cell. The first such state should be at 1/4-filling. This physics arises naturally from a momentum-space formulation of the interactions which will result in 4-poles of the Green function, each corresponding to the four insulating states possible.

We now introduce the Hatsugai–Kohmoto (HK) interaction[70–72] into a general QSH Hamiltonian,

$$H = \sum_{\mathbf{k},\sigma} \left[ (\varepsilon_{+,\mathbf{k},\sigma} - \mu) n_{+,\mathbf{k},\sigma} + (\varepsilon_{-,\mathbf{k},\sigma} - \mu) n_{-,\mathbf{k},\sigma} \right]$$
$$+ U \sum_{\mathbf{k}} (n_{+,\mathbf{k},\uparrow} n_{+,\mathbf{k},\downarrow} + n_{-,\mathbf{k},\uparrow} n_{-,\mathbf{k},\downarrow}). \tag{3}$$

Without loss of generality, we use the dispersions from the BHZ model (see Methods) setting $M = 1$ as an example. This interaction introduces Mottness by tethering double occupancy to k-space rather than the usual real space as in the well-known Hubbard model. As we will show, this model yields physics for strong interactions consistent with the Hubbard model. The reason for this consilience[72] is that both models break the underlying $\mathbb{Z}_2$ (distinct from the classification scheme for topological insulators) symmetry of the non-interacting Fermi surface[73]. As the interaction commutes with the kinetic term, the original non-interacting wave function is untouched and momentum **k** remains a good quantum number. Therefore, it makes sense to extract the Chern number from an integration over the Brillouin zone. The interacting Green function can be written down analytically[67,71] as

$$G_{\pm,\mathbf{k},\sigma}(\omega) = \frac{1 - \langle n_{\pm,\mathbf{k}\bar\sigma} \rangle}{\omega + \mu - \varepsilon_{\pm,\mathbf{k},\sigma}} + \frac{\langle n_{\pm,\mathbf{k}\bar\sigma} \rangle}{\omega + \mu - (\varepsilon_{\pm,\mathbf{k},\sigma} + U)}. \tag{4}$$

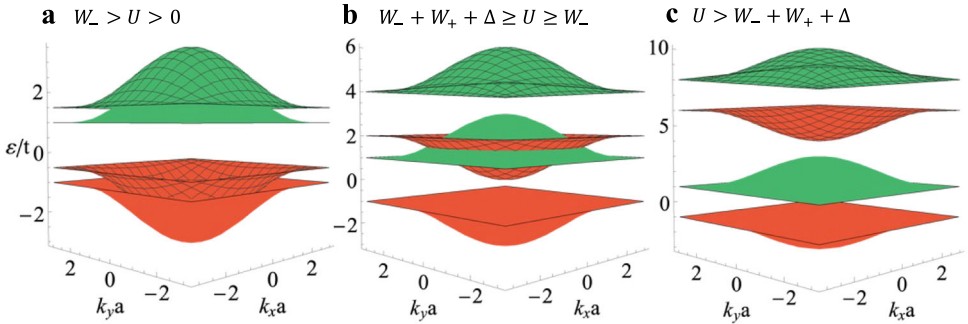

**Fig. 4 | Band structure for BHZ-HK model in Eq. (3) with $M = 1$.** Different phases emerge as $U$ increases: **a** 1/2-filled QSH insulator for $W_{0-} > U > 0$ ($U = 0.5$), **b** 1/4-filled TMI and 1/2-filled metal for $W_{0-} + W_{0+} + \Delta_0 \geq U \geq W_{0-}$ ($U = 3$), and **c** 1/4-filled QSH Mott insulator and 1/2-filled topologically trivial Mott insulator for $U > W_{0-} + W_{0+} + \Delta_0$ ($U = 7$). The red (or green) color represents $C_s = 1$ (or $-1$). The unmeshed (meshed) band consists of only singly (doubly) occupied states. It is the splitting of these bands by the interaction that gives rise to the Mott-derived topological physics.

The Green function immediately reveals the effect of the correlations. The non-interacting lower and upper bands which were degenerate for spin-up and spin-down electrons split into singly and doubly occupied sub-bands as a result of Mottness. In the following, we use the abbreviation LSB and LDB for lower singly and doubly occupied sub-bands respectively, and likewise USB and UDB for the upper bands. The energy of the LSB and USB remains at the non-interacting value, while the LDB and UDB move up by a value equal to $U$. For a large enough $U$, the quarter-filled system emerges as an insulator with a filled LSB. This physics falls out naturally from the HK model because of the 4-pole structure of the Green function.

Since the interaction mixes the spin channels, leading to a huge degeneracy ($d = 2^{N_c}$) in the ground state ($N_c$ is the number of unit cells), we need to average over all degenerate ground states[74] to rigorously calculate the spin Chern number: $\bar{C}_s = \bar{C}_\uparrow - \bar{C}_\downarrow$. For each spin, the contribution is

$$\bar{C}_\sigma = \frac{1}{d} \sum_{\Omega=1}^{d} \frac{1}{2\pi} \int d^2 k f_{xy,\sigma} \langle \Omega | n_{\mathbf{k},\sigma} | \Omega \rangle, \tag{5}$$

where $f_{xy,\sigma}$ is the normal Berry curvature defined with Bloch wave function[75] because $k$ remains a good quantum number in the HK model, and $(1/2\pi) \int d^2 k f_{xy,\sigma} = C_{0\sigma}$. When $U = 0$, $\langle \Omega | n_{\mathbf{k},\sigma} | \Omega \rangle = 1$ below the chemical potential. When $U$ is finite, $\langle \Omega | n_{\mathbf{k},\sigma} | \Omega \rangle$ can be 0 or 1. We can conduct the average first for Eq. (5). When $U$ is large enough to fully separate the singly and doubly occupied bands, $(1/d) \sum_{\Omega=1}^{d} \langle \Omega | n_{\mathbf{k},\sigma} | \Omega \rangle = \langle n_{\mathbf{k},\sigma} \rangle = \langle n_\sigma \rangle = 1/2$. Then Eq. (5) becomes

$$\bar{C}_\sigma = \langle n_\sigma \rangle \frac{1}{2\pi} \int d^2 k f_{xy,\sigma} = \langle n_\sigma \rangle C_{0\sigma} = \frac{C_{0\sigma}}{2}. \tag{6}$$

Thus, the spin Chern number $C_s = C_{0s}/2$ (we will drop the average bar symbol in the following text.). This result demonstrates that each momentum state is equivalently occupied by half spin-up and half spin-down electrons on average. Similarly, the LDB has the same $C_s$, while the USB and UDB have the opposite $C_s$. In short, the strongly correlated quarter-filled system becomes a Mott insulator with a spin Chern number $C_s = C_{0s}/2$ should the interaction exceed the bandwidth.

To visualize how this phase emerges, we plot the band structure in Fig. 4 for varying $U$. With $M = 1$, the bandwidth for the lower and upper BHZ bands is $W_{0+(-)} = 2$ and $\Delta_0 = 2$ is the topological gap. The non-interacting lower band has $C_{0s} = 2$, while the upper band carries the opposite spin Chern number. We separate the non-interacting lower and upper bands into LSB (red-unmeshed), LDB (red-meshed), USB (green-unmeshed) and UDB (green-meshed). As derived above, the red and green sub-bands have the spin Chern number $C_s = 1$ and $-1$ respectively. Turning on the interaction causes the doubly occupied

sub-bands to increase in energy while the singly occupied sub-bands remain unchanged. For small interactions $W_{0-} > U > 0$ (Fig. 4a), the band structure only slightly departs from the non-interacting case. As $U$ increases to $W_{0+} + W_{0-} + \Delta_0 \geq U \geq W_-$ (Fig. 4b), the same-color sub-bands fully separate, leading to a gap opening at quarter-filling. Then both the 1/4- and 3/4-filled systems become a TMI with a spin Chern number $C_s = 1$, while the 1/2-filled case becomes a conductor. Upon further increasing $U$ to $U > W_{0+} + W_{0-} + \Delta_0$ (Fig. 4c), the 1/2-filled state becomes a topologically trivial Mott insulator. All the while, the QSH Mott insulator at 1/4- and 3/4-fillings persists with a gap equal to $\Delta$. For a different $M$, the intermediate panel b may change, while panel c is always valid for a large enough $U$. This indicates that generally in the presence of strong interactions, the system becomes a QSH Mott insulator at 1/4- and 3/4-filling with spin Chern number $C_s = C_{0s}/2$ and a trivial Mott insulator at 1/2-filling.

As we compute in the Supplement, the spin susceptibility for the HK model diverges at $T = 0$ indicating that the HK model is unstable to ferromagnetic order in this limit. Note that this conclusion applies to a general QSH Hamiltonian (not only to the BHZ model) with HK interactions. This result is consistent with the divergence of the spin-susceptibility of the flat-band KM-Hubbard models. Ultimately this means that the 1/2-filled QSH effect would give rise to a 1/4-filled QAH effect at $T = 0$.

To summarize, this simple model offers a way of understanding why 1/4-filling is special in the Hubbard model. Note the agreement with the Hubbard simulations is non-trivial because momentum mixing is not present in HK model but is in the Hubbard model. Hence, the agreement demonstrates that it is the ultimate 4-pole structure of the underlying single-particle Green function that dictates the physics. As we have shown previously[72,76], the HK model represents a fixed point in which no short-range repulsions are relevant not even Hubbard interactions. Hence, the HK model is the fixed point for Mott physics. Possible ferromagnetism at $T = 0$ would eventually turn the QSH effect into the QAH effect. Hence, as a result of interactions, the QAH effect appears as the symmetry-broken phase of the QSH effect much the way antiferromagnetism is the low-temperature symmetry-broken phase of a Mott insulator. Equal drivers of this spontaneous symmetry breaking are consistency with the LSM theorem and the restriction that the Chern number must be an integer. As is evident at 1/4-filling, the QAH always dominates as the symmetry-broken ground state. This is the primary conclusion of this work.

In a previous exact diagonalization study[77] on a strict flat band model with Hubbard interactions and spin-orbit interaction, it was noticed that ferromagnetism emerged at 1/4-filling. This result can be viewed as a special case of HK physics because in the strict flat-band limit of the model studied, any value of $U$ will necessarily prohibit double occupancy thereby producing a gapped state. The HK result is

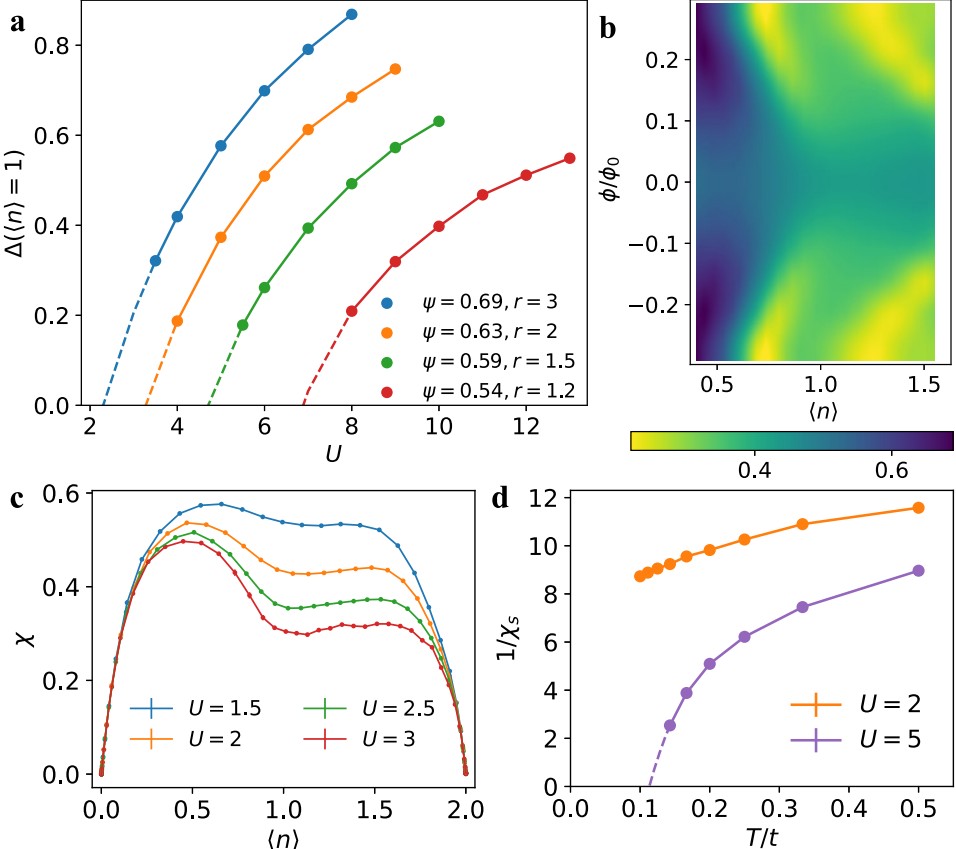

**Fig. 5 | Gap opening and non-trivial topology at quarter-filling of the generalized KM-HH model. a** Estimated gap of the generalized KM-Hubbard model at quarter-filling as a function of the interaction strength $U$ and the hopping phase $\psi$. These results are obtained from DCA simulations at a temperature of $\beta = 20/t$. **b** The DQMC compressibility for the KM-HH model at $U = 2t$, $\beta = 8$ around quarter-filling.

**c** The zero-field DQMC compressibility for the KM-Hubbard model at various $U$ and $\beta = 8$ as a function of the density. **d** The inverse temperature-dependent spin susceptibility from DQMC for KM-Hubbard model at $U = 2t$ and $5t$. **b**–**d** fix $\psi = 0.63$. All DQMC simulations are done on a $6 \times 6 \times 2$ cluster while the DCA simulations are on a $2 \times 2 \times 2$ cluster.

more general than this result as the gap persists even when the bands disperse.

## Gap opening

In the previous sections, we have shown that both simulations on the Hubbard model and analytical calculations on the HK model indicate the emergence of non-trivial topology at 1/4-filling driven by strong correlations. In the non-interacting case, the topology appears with a bulk gap. In the strongly correlated case, however, this is not necessarily true. While a gap opens in the HK model as long as $U$ exceeds the total bandwidth, the precise condition for opening a gap in the Hubbard model is much more subtle because of the dynamical mixing between the bands. In the Hubbard case, the interaction strength needs to exceed a critical value ($U_c^{topo} \gg W_{0-}$) to induce the topology at 1/4-filling and a separate critical value ($U_c^{gap}$) to open a gap. In general, we find $U_c^{topo} < U_c^{gap}$.

From the dip of the high-temperature compressibility computed by DQMC, we can tell roughly when the non-trivial topology appears and hence we are able to extract $U_c^{topo}$. However, to access the gap information, one has to explore much lower temperatures. This cannot be done by DQMC as we are restricted by the Fermion sign problem and finite-size effects (see Supplement for details). To address this problem, we resort to DCA[78–81]. We computed the value of the gap defined as

$$\Delta(\langle n \rangle = 1) = \mu(\langle n \rangle = 1.01) - \mu(\langle n \rangle = 0.99), \quad (7)$$

in the vicinity of the quarter-filled state in the generalized KM-Hubbard model using DCA on a $2 \times 2 \times 2$ cluster at low temperature $\beta = 20/t$. To make contact with previous work on flat-band systems, we define the ratio $r = \Delta_0/W_{0-}$ and study the evolution of the gap as a function of the complex hopping phase, $\psi$ in the generalized KM model (fixing $t' = 0.3$). The results are summarized in Fig. 5a. We only plot the data when $\Delta(\langle n \rangle = 1) \gtrsim 0.2$ because $\Delta(\langle n \rangle = 1)$ by definition remains a small value even when the state is metallic and obtain the $U_c^{gap}$ by extrapolation to zero gap. In all cases, $\Delta(\langle n \rangle = 1)$ is significantly smaller than $\Delta_0$ even when $U > W_{0-} + W_{0+} + \Delta_0$ ($\approx 6$). As the band becomes more dispersive ($\psi$ decreases, or $r$ decreases), $\Delta(\langle n \rangle = 1)$ reduces and $U_c^{gap}$ grows as shown in Fig. 5a ($\Delta_0 = 2$ for all cases). Now we consider the relation between $U_c^{gap}$ and $U_c^{topo}$. Take $\psi = 0.63$ as an example ($\Delta_0 = 2$, $W_{0-} = 1$). Already at $U = 2t$, the corresponding KM-HH model shows QAH topology at $\beta = 8$ in Fig. 5b, while $U_c^{gap} \approx 3.25$. The DQMC compressibility at zero field is shown in Fig. 5c at various $U < U_c^{gap}$. It exhibits that a dip at $\langle n \rangle = 1$ starts to develop (and therefore the topological magnetic response) at a smaller $U$ before the gap actually opens (see Supplement for a benchmark between DCA and DQMC). This supports a topological Mott semimetal (TMSM). In Fig. 5d, we find that for the semi-metallic state, the inverse spin susceptibility $1/\chi_s$ decreases slowly with temperature and is unlikely to reach 0 at finite temperatures, while for the insulating state, the $1/\chi_s$ drops much sharper with temperature so that its extrapolation supports a finite-temperature transition. We conclude that while the Chern numbers remain the same in the TMSM and insulating QAH phases, the TMSM phase lacks a gap and ferromagnetism as well.

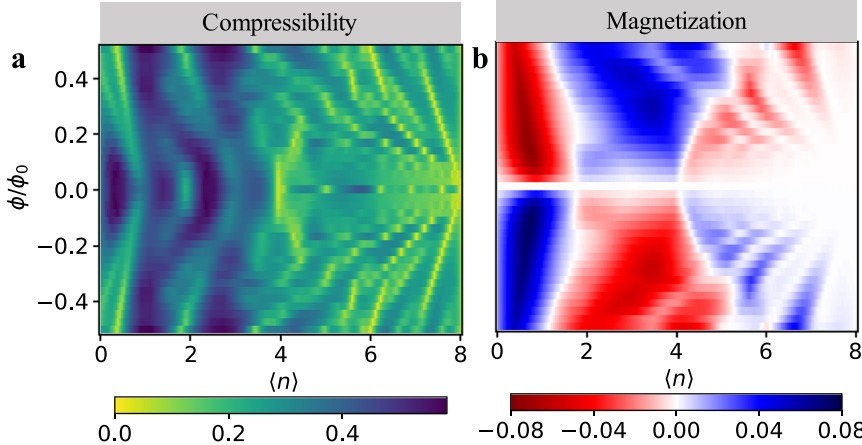

**Fig. 6 | Compressibility and magnetization of the bilayer KM-HH model.** DQMC results for the bilayer KM-HH model at $U = 1.5t$, $\beta = 12/t$ with an interlayer hopping $t_\perp = 0.3t$ and voltage difference between the two layers of $V = 0.4t$. **a**, **b** show the compressibility and magnetization, respectively, as a function of magnetic flux and electron density.

In Fig. 5a, at any finite value of $r > 1$, $U$ must exceed a critical value $U_c^{\mathrm{gap}}$ for the gap to form. In general, $U_c^{\mathrm{gap}}$ increases as $r$ decreases. When $r = 1$ ($\psi = 0.5$), we find the gap barely opens for $U < 12$ ($\Delta(\langle n \rangle = 1, U = 12) = 0.2$). This situation is directly applicable to the BHZ-Hubbard model because in the topologically relevant region ($-2 < M < 2$), $r \leq 1$. At $M = 1$ (which means $r = 1$), $\Delta(\langle n \rangle = 1, U = 12) = 0.22$. However, we already observe the emergence of non-trivial topology at 1/4- and 3/4-filling in Fig. 3 at $U = 8$ and even $U = 6$ (see Supplement). Thus, such phases with non-trivial topology in actuality are semi-metals. Similar to the KM-Hubbard model, the spin susceptibility for these semi-metallic states only shows a soft peak and their temperature evolution does not support a finite-temperature transition (see Supplement). In such a system, the Hall conductance will have finite temperature corrections[36] and hence deviate from the value dictated by the Chern number. This observation is in contrast with the recent exact diagonalization study on the BHZ-Hubbard model (with system size up to $3 \times 4$) which observes gap opening and ferromagnetic order for $U > 4$. As we show in the Supplement, the finite size effects are sizeable for the cluster size used in this study.

**Experimental realization**

While the interactions in ultracold atoms in optical traps[82] can be adjusted to mimic the physics here, the most obvious synergy is with the moiré TMD experiments[18,19,32] discussed previously. Our DQMC simulation result in Fig. 1 for the flat-band KM-HH model is consistent with this experiment in the existence of QAH and QSH at 1/4- and 1/2-filling, respectively.

However, we cannot make direct contact with the observation of valley coherence[32] within a single-layer KM model in which spin-valley locking obtains. Note relaxing the spin-valley locking constraint of the KM model by reversing the spins in one of the bands relative to the other, as indicated in the experiment[32] (see Supplement), would lead to a contradiction with a non-zero Chern number per spin in the band insulator limit. That is, the moiré band structure of AB stacked MoTe$_2$/WSe$_2$ bilayer can not be captured by a strict four-band model such as the KM model. The remedy is to construct an eight-band model (details in Supplement) consisting of two copies of the KM model, one for each layer with an effective voltage difference between the layers. For completeness, we recomputed the compressibility for the bilayer flat-band KM-HH model at an intermediate $U = 1.5t$. Clearly shown in Fig. 6a is the QAH at $\langle n \rangle = 1$, the QSH at $\langle n \rangle = 2$ and $\langle n \rangle = 4$. Besides, there is also an emergent QAH state at $\langle n \rangle = 3$. This prediction has been confirmed in a recent experiment on a moiré TMD material[83].

The accompanying magnetization in Fig. 6b is also consistent with these assignments. Within the eight-band model, spin polarization requires layer coherence because the interaction does not commute with the interlayer hopping and since the same spin is assigned to different valleys in each layer, layer coherence necessarily entails valley coherence. Hence, a simple two-layer extension of our results is sufficient to account for the QAH effect in TMD moiré systems. This reasoning motivates first-principle calculations to determine how the 8-band model should be tailored to apply to specific moiré materials.

## Discussion

Interactions play a non-trivial role in topology in two distinct ways. First, they lead to a TMSM/TMI with a high-temperature QSH effect characterized by a spin Chern number of $C_s = 1$ at quarter filling in the interacting BHZ and KM models. We use the term "Mott" because it is the interactions that lead to a lifting-up of the doubly occupied sector thereby exposing the topology of the 1/4-filled band. The resultant $C_s = 1$ poses a problem as this number must be even for a non-degenerate ground state[84] with time-reversal symmetry. The resolution of this dilemma lies in the divergence of the spin susceptibility in the HK model at zero temperature and in the Hubbard model at finite temperature. Both of these indicate a possible spontaneous ferromagnetic phase at $T = 0$. The onset of ferromagnetism results with a unit Chern number indicative of the QAH effect and would offer a route around the LSM restriction[33,34] that a unique featureless gapped ground state is impossible with an odd number of fermions per unit cell. Consequently, our results point to a fundamental reason why the QSH effect at high temperatures must resort to the QAH effect as temperature decreases if a gap opens. Namely, while at high temperatures, a paramagnetic symmetry-unbroken state obtains, for the ground state to be unique, the symmetry must be spontaneously broken. We refer to this onset of the symmetry-broken state as a consequence of topological Mottness[85], in direct analogy with the traditional Mott state which has an antiferromagnetic ground state. Therefore, we argue that the 1/4-filled state is a TMI or TMSM. In analogy with the traditional Mott insulator with an antiferromagnetic ground state, a TMI exhibits the QSH phase which turns into the symmetry-broken QAH at low temperature as illustrated in Table 1. A TMI is qualitatively distinct from the fractional topological insulators[4,10,59–61] driven by at least nearest-neighbor interactions. The fractional topological insulator usually consists of two decoupled fractional Chern insulators with opposite spins. However, in the TMI, spin-up and -down electrons are correlated to form the inseparable singly occupied states giving rise to the high-temperature QSH feature and a QAH ground state. Second, we showed that in the flat-band limit, a high-

**Table 1 | Comparison between four different insulators**

| Classification scheme | | | | | |
|---|---|---|---|---|---|
| | | **TMI** | **TI** | **MI** | **BI** |
| Filling | | 1/4 | 1/2 | 1/2 | 1 |
| Topology | $T > T_{th}$ | QSH | QSH | Trivial | Trivial |
| | | ($C_s = 1$) | ($C_s = 2$) | | |
| | $T = 0$ | QAH | QSH | Trivial | Trivial |
| | | ($C = \pm 1$) | ($C_s = 2$) | | |
| Magnetism | $T > T_{th}$ | PM | PM | PM | PM |
| | $T = 0$ | FM | PM | AFM | PM |

$T_{th}$ is a threshold temperature above which the symmetry is maintained. Here we assume two atoms per unit cell for all cases, and $U$ is sufficiently large for Mott physics to dominate in a TMI and MI.

*TMI* topological Mott insulator, *TI* topological insulator, *MI* Mott insulator, *BI* band insulator, *PM* paramagnetism, *FM* ferromagnetism, *AFM* antiferromagnetism.

temperature QAH state exists also at 1/4-filling with an intermediate $U$. In the double-layer extension of this model, the QAH state exhibits valley coherence as is seen experimentally in moiré TMD materials[32].

## Methods

### The model

All QSH models are based on Hamiltonians of the form,

$$H_{QSH} = \sum_{\mathbf{k}} \Phi^\dagger(\mathbf{k}) \begin{pmatrix} h_{QAH}(\mathbf{k}) & 0 \\ 0 & h^*_{QAH}(-\mathbf{k}) \end{pmatrix} \Phi(\mathbf{k}), \quad (8)$$

where $\Phi^\dagger = \{c^\dagger_{O_1,\uparrow}, c^\dagger_{O_2,\uparrow}, c^\dagger_{O_1,\downarrow}, c^\dagger_{O_2,\downarrow}\}$ is a four-component spinor, where $O_{1/2}$ stands for different orbitals or sub-lattices, respectively. Eq. (8) means that the spin-up and spin-down electrons are described by a QAH Hamiltonian $h_{QAH}(\mathbf{k}) = h_a(\mathbf{k})\tau^a$ ($\tau^a$ is the Pauli matrix for orbital/sublattice space) and its TR conjugate counterpart $h^*_{QAH}(-\mathbf{k})$ with opposite chirality. As a result, the system is TR invariant and the half-filled case can be a topologically trivial and non-trivial insulator, categorized by a $\mathbb{Z}_2$ invariant or the spin Chern number $C_s$ if $\hat{S}_z$ is conserved. As a consequence, any ferromagnetism here will be of the Ising type rather than $U(4)$ as in the BM model[49,53]. To introduce Hubbard on-site interactions, we need to resort to a real-space representation of the QSH model. For concreteness, consider the generalized KM model[18] in the honeycomb lattice under an external magnetic field, namely the KM-Hubbard-Hofstadter (KM-HH) model:

$$\begin{aligned} H = \sum_{\mathbf{ij}\sigma} t_{\mathbf{i,j}} \exp(i\phi_{\mathbf{i,j}}) c^\dagger_{\mathbf{i}\sigma} c^\dagger_{\mathbf{j}\sigma} - \mu \sum_{\mathbf{i},\sigma} n_{\mathbf{i}\sigma} \\ + \lambda_\nu \left( \sum_{\mathbf{i}\in A,\sigma} n_{\mathbf{i}\sigma} - \sum_{\mathbf{i}\in B,\sigma} n_{\mathbf{i}\sigma} \right) + U \sum_{\mathbf{i}} \left( n_{\mathbf{i}\uparrow} - \frac{1}{2} \right) \left( n_{\mathbf{i}\downarrow} - \frac{1}{2} \right), \end{aligned} \quad (9)$$

where $t_{\mathbf{i,j}}$ contains the nearest-neighbor hopping $t = 1$ (as the energy scale) and next-nearest-neighbor hopping $t'e^{\pm i\psi\sigma}$ as the spin-orbit coupling with $\pm i\psi$ following the convention in the Haldane model[17]. If we set $\psi = 0.5$ (in the unit of $\pi$), the hopping term reduces to the original KM model[1,2]. $\lambda_\nu$ is the sub-lattice potential difference. For simplicity, we fix $\lambda_\nu = 0$ for this study. Non-trivial topology arises as long as $t' \neq 0$, $\psi \neq 0, 1$. The phase factor $\exp(i\phi_{\mathbf{i,j}})$ which arises from the standard Peierls substitution contains the effect of the external magnetic field, which is introduced to measure the magnetic response of the incompressible states at high temperature to determine the topology. Here $\phi_{\mathbf{i,j}} = (2\pi/\Phi_0) \int_{r_\mathbf{i}}^{r_\mathbf{j}} \mathbf{A} \cdot d\mathbf{l}$, where $\Phi_0 = e/h$ is the magnetic flux quantum, the vector potential $\mathbf{A} = (x\hat{y} - y\hat{x})B/2$ (symmetric gauge), and the integration is along a straight-line path.

The other model we study is the BHZ-Hofstadter-Hubbard (BHZ-HH) model:

$$\begin{aligned} H = t \sum_{\mathbf{i},\sigma} \Bigg[ \exp(i\phi_{\mathbf{i,i}+\hat{x}}) c^\dagger_{\mathbf{i},\sigma} \frac{\tau_z - i\sigma\tau_x}{2} c^\dagger_{\mathbf{i}+\hat{x},\sigma} \\ + \exp(i\phi_{\mathbf{i,i}+\hat{y}}) c^\dagger_{\mathbf{i},\sigma} \frac{\tau_z - i\tau_y}{2} c_{\mathbf{i}+\hat{y},\sigma} + \text{h.c.} \Bigg] - \mu \sum_{\mathbf{i},\sigma} n_{\mathbf{i},\sigma} \\ + M \sum_{\mathbf{i},\sigma} c^\dagger_{\mathbf{i},\sigma} \tau_z c_{\mathbf{i},\sigma} + U \sum_{\mathbf{i}\alpha} \left( n_{\mathbf{i}\alpha\uparrow} - \frac{1}{2} \right) \left( n_{\mathbf{i}\alpha\downarrow} - \frac{1}{2} \right), \end{aligned} \quad (10)$$

where $t = 1$ (energy scale), $\tau_a$ is the Pauli matrix in the orbital basis and $\alpha$ is the orbital index. Non-trivial topology arises as long as $|M| < 2$.

At zero field, a general QSH Hamiltonian can be diagonalized into

$$H_{QSH} = \sum_{\mathbf{k},\sigma} \left[ (\varepsilon_{+,\mathbf{k},\sigma} - \mu) n_{+,\mathbf{k},\sigma} + (\varepsilon_{-,\mathbf{k},\sigma} - \mu) n_{-,\mathbf{k},\sigma} \right], \quad (11)$$

where $\mu$ is the chemical potential and

$$\varepsilon_{\pm,\mathbf{k},\sigma} = h_{0,\sigma}(\mathbf{k}) \pm \sqrt{h^2_{x,\sigma}(\mathbf{k}) + h^2_{y,\sigma}(\mathbf{k}) + h^2_{z,\sigma}(\mathbf{k})} \quad (12)$$

represents the upper ($+$) and lower ($-$) bands for each spin. In the BHZ[3] model,

$$\begin{aligned} h_{0,\sigma}(\mathbf{k}) = 0, \quad h_{x,\sigma}(\mathbf{k}) = \sigma t \sin(k_x), \\ h_{y,\sigma}(\mathbf{k}) = t \sin(k_y), \quad h_{z,\sigma}(\mathbf{k}) = M + t\cos(k_x) + t\cos(k_y), \end{aligned} \quad (13)$$

The spin-up and -down electrons have the same dispersion but different wave functions with opposite chirality. For $2 > M > 0$ (or $-2 < M < 0$), the half-filled system is a QSH insulator[3] with $C_{0s} = 2$ (or $-2$) related to the spin Hall conductance[3,9].

### Numerical simulations

We use the DQMC method[86–88] to simulate the KM-HH and BHZ-HH models on an $N_{site} = 6 \times 6 \times 2$ cluster (two sublattices or orbitals per unit cell) with modified periodic boundary conditions[89]. A single-valued wave function requires the flux quantization condition $\Phi/\Phi_0 = n_f/N_c$ (with $n_f$ an integer). We also use DCA to calculate the charge gap at low temperatures on a $N_{site} = 2 \times 2 \times 2$ cluster of the KM-MM model. The DCA represents the infinite lattice in the thermodynamic limit by a finite cluster embedded in a self-consistent dynamical mean field. It has a much milder finite-size effect and Fermion sign problem. The technical details of these two methods are provided in the Supplement.

## Data availability

The DQMC and DCA data generated in this study have been deposited in the Zenodo under the accession code https://doi.org/10.5281/zenodo.8275156.

## Code availability

The DQMC code used for this project can be obtained at https://doi.org/10.5281/zenodo.8275145. The DCA code for this study can be obtained at https://doi.org/10.5281/zenodo.8275154.

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

## Acknowledgements

We thank Taylor L. Hughes, Edwin W. Huang, Kin Fai Mak and Kam Tuen Law, Cristian Batista, Thomas Maier, Charlie Kane and Barry Bradlyn for useful discussions. We also thank P. Armitage for help with the pithy title. This work was supported by the Center for Quantum Sensing and Quantum Materials, a DOE Energy Frontier Research Center, grant DE-SC0021238 (P.M., B.E.F., and P.W.P.). P.W.P. also acknowledges NSF DMR-2111379 for partial funding of the HK work, which led to these results. The DQMC calculation of this work used the Advanced Cyberinfrastructure Coordination Ecosystem: Services & Support (ACCESS) Expanse supercomputer through the research allocation TG-PHY220042, which is supported by National Science Foundation grant number ACI-1548562[90].

## Author contributions

P.M. performed the DQMC and DCA calculations on the Hubbard model, analyzed the data, and carried out the analytic calculations on the HK model; J.Z. provided the method to calculate the spin Chern number in HK model and analyzed the data; B.E.F. provided information on experimental realizations; P.W.P. supervised the project; P.M and P.W.P. wrote the paper with input from all authors.

## Competing interests

The authors declare no competing interests.
