## [Peer Review File · Nature Communications]

REVIEWER COMMENTS

Reviewer #1 (Remarks to the Author):

The authors use (unbiased) determinant quantum Monte Carlo simulations to study interacting topological models. In particular, their primary focus is on the Bernevig-Hughes-Zhang-Hofstadter-Hubbard (BHZ-HH), analyzing the behavior of the charge compressibility, the spin susceptibility, and the magnetization. They do so by inserting a probe magnetic field, via Peierls substitution, which is then used to quantify the topological invariants as the (spin) Chern number. Subsequently, they support their results on the approximate model with local-in-momentum interactions, the Hatsugai-Kohmoto model. The rationale is, according to them, "this model yields physics for strong interactions consistent with the Hubbard model ." Finally, to extend the scope of the work, they study the Kane-Mele-Hofstadter-Hubbard model, a variant in which the lower band is flatter, and a second variant, a bilayer Kane-Mele-Hofstadter-Hubbard with added energy offset between the two layers. I acknowledge the amount of work put in here by them. There is an overwhelming amount of information in the text, and I thank the authors for putting Table I to summarize things.

The main result is that in the original model, BHZ-HH, while the physics at half-filling of a quantum spin Hall ground-state gives way to a trivial Mott insulator with increasing Hubbard interactions U (harboring spin order), they claim that at quarter-filling (or three-quarter filling) the trivial metal gives way to a non-trivial topological state with increasing U , even at finite (and quite large temperatures).

I think one of the most apparent "issues" with the work is that they are limited to large temperatures T owing to the manifestation of the sign problem. However, according to their logic, this is not an issue because, at finite T , one already has the manifestation of the quantum spin Hall (QSH) effect, a topologically non-trivial phase. Lowering T , their results are *not* directly connected to the recently published ones coming from exact diagonalization [Ref. 35], which exhibits that the ground state should be a quantum anomalous Hall (QAH) effect instead. These indeed come from small cluster calculations but combined with the arguments of Lieb-Schultz-Mattis theorem and its extension [Refs. 1 and 2] it shows that there should be a thermal transition at T_{th} to compatibilize their results, as they roughly propose in the text and more explicitly in the cartoon Figure 4. They cannot access such transition.

From what I could understand, no *direct* evidence in their results points to the connection between the QSH at $T > T_{th}$ and the QAH at $T = 0$ obtained on the same model using ED. The Hatsugai-Kohmoto model is used to provide support. Here the spin Chern number at quarter-filling gives $C_s = 1$, half of the non-interacting value at the same value of M at half-filling (another QSH).

These calculations are obviously done at $T = 0$. However, they are already incompatible with the $T = 0$, ED results, which give a $C_s = 0$ (and Chern number $C = 1$). So one just hopes there should be a transition temperature to make these results (DQMC at large T and ED at $T = 0$) compatible.

Finally, if the authors are confident that their (unbiased) results unequivocally point out to a QSH effect at large T , what is the minimal temperature one needs to set in to see their appearance? In other words, the upper line in Fig. 4, separating a metal from the QSH, is a transition or a crossover? This is not a problem for DQMC, obviously.

In summary, I don't think the authors are wrong, but there are some gaps in the logic to make their results compatible with other exact ones. I would be keen to hear from the authors before making an assessment about recommendation for publication.

Some points:

The $U = 0$ calculation of the BHZ-HH-(TRI) model can be done at any temperature in the DQMC (one field configuration suffices to obtain the physical properties). Have the authors tested that in the large β limit, one recovers the known results for this model? In particular, the slopes of the compressibility to give the spin Chern number? I suspect the features in Fig.1(a-d) to be sharper. I understand the motivation in the main text of using the same T as for the finite U calculations in Fig.1(e-h), but including it in the SM might be interesting.

The $T = 0$ phase at quarter filling should harbor a ferromagnetic ground-state. Large spin fluctuations are seen at this density for the $T > 0$ DQMC calculation. Owing to the Mermin-Wagner theorem, do they expect that there could be instead a quasi-long-range order at a finite temperature a la Kosterlitz-Thouless that defines T_{th} ? Or is this independent?

When the authors study the Kane-Mele Hubbard model, they show in Fig. S1(b) a mapping of the sign vs. the density. From what I understand from Refs. PHYSICAL REVIEW B 84, 205121 (2011) and PHYSICAL REVIEW B 85, 115132 (2012), at half-filling in such a time-reversal model, one can show that the sign problem is absent. However, this figure shows that although $\langle S \rangle$ is peaked at this density, it does not reach 1. What gives?

Minor points:

Reproducibility is facilitated by knowing the details of the simulations. That is, what is the imaginary-time discretization used? The type of Hubbard-Stratonovich transformation? The typical sampling?

Sentences such as "The results are striking." on page 6 bear no place in scientific texts. The community comes up with those if appropriate.

Scultz -> Schultz in the abstract.

Caption of Fig. 4 needs to explicitly state this is a cartoon or schematic representation of the expected physics, since none of it is based on actual data.

Reviewer #2 (Remarks to the Author):

Motivated by recent experiments moire TMD bilayers, the authors study a model with a QSH ground state at half-filling that leads to a QAH state at quarter filling. The QSH state is smoothly connected to a non-interacting state and provided that the interaction does not exceed the gap remains robust. The main surprise, according to the authors, is the emergence of a ferromagnetic quarter-filling QAH state. The work is interesting and the numerical analysis is thorough. However, I disagree with the framing of the results as surprising or qualitatively novel. The emergence of QAH states in models of narrow topological bands has been extensively studied over the past few years in the context of graphene moire systems and the main physics is reasonably well-understood as detailed below. Without placing the results in the context of recent findings in the field, it is difficult to assess its novelty. Thus, I do not recommend the manuscript for publication in its current form but am open to reconsider following suitable revisions. Let me elaborate below my concerns.

1. The model discussed by the authors for QSH (the BHZ model or even the Kane-Mele model) describes a 2D TI if only spinful time-reversal symmetry is preserved and a QSH insulator if $U(1)$ S_z symmetry is preserved. In limit where the interaction is smaller than the gap to the remote bands, the model describes a pair of opposite Chern bands labelled by opposite S_z which map to each other under time-reversal and are subject to strong interactions. This is identical to the problem considered in the context of graphene moire systems where $U(1)$ valley plays the role of S_z (see for example Bultinck et. al. PRL 124, 166601 (2020), Zhang et. al. PRB 99, 075127 (2019)). This problem is well-understood as a generalization of quantum Hall ferromagnetism where exact results can be obtained in the flat band limit (Bultinck et. al. PRX 10, 031034 (2020), Lian et. al. 103, 205414 (2021)). This picture has by now been verified by exact diagonalization (Xie et. al. PRB 103, 205416 (2021)), DMRG (Soejima et. al. 102, 205111 PRB (2020)), and QMC (Hofmann et. al. PRX 12, 011061 (2022)). The authors need to comment on the similarity/difference between the two problems and whether there is a reason to expect any qualitatively new aspect in the model they considered compared to the ones studied in graphene systems?

2. In the discussion of the HK model, it is unclear what conclusions drawn by the authors are general and which ones are specific to the model. For instance, the authors mention there is a very large degeneracy (2^N) for the ground state manifold whereas the general expectation for a ferromagnet is $N+1$ for $SU(2)$ spin symmetry and 2 for $U(1)$ spin symmetry. Is this a consequence of the specific form of the interaction (HK) that is used?

RESPONSE TO REVIEWER #1

The Reviewer wrote: The authors use (unbiased) determinant quantum Monte Carlo simulations to study interacting topological models. In particular, their primary focus is on the Bernevig-Hughes-Zhang-Hofstadter-Hubbard (BHZ-HH), analyzing the behavior of the charge compressibility, the spin susceptibility, and the magnetization. They do so by inserting a probe magnetic field, via Peierls substitution, which is then used to quantify the topological invariants as the (spin) Chern number. Subsequently, they support their results on the approximate model with local-in-momentum interactions, the Hatsugai-Kohmoto model. The rationale is, according to them, "this model yields physics for strong interactions consistent with the Hubbard model ." Finally, to extend the scope of the work, they study the Kane-Mele-Hofstadter-Hubbard model, a variant in which the lower band is flatter, and a second variant, a bilayer Kane-Mele-Hofstadter-Hubbard with added energy offset between the two layers. I acknowledge the amount of work put in here by them. There is an overwhelming amount of information in the text, and I thank the authors for putting Table I to summarize things.

Our response: Many thanks to the referee for taking the time to appreciate the details of the simulations and the HK model.

The Reviewer wrote: The main result is that in the original model, BHZ-HH, while the physics at half-filling of a quantum spin Hall ground-state gives way to a trivial Mott insulator with increasing Hubbard interactions U (harboring spin order), they claim that at quarter-filling (or three-quarter filling) the trivial metal gives way to a non-trivial topological state with increasing U , even at finite (and quite large temperatures).

Our response: We thank the referee for summarizing our work and acknowledging our effort.

The Reviewer wrote: I think one of the most apparent "issues" with the work is that they are limited to large temperatures T owing to the manifestation of the sign problem. However, according to their logic, this is not an issue because, at finite T , one already has the manifestation of the quantum spin Hall (QSH) effect, a topologically non-trivial phase. Lowering T , their results are *not* directly connected to the recently published ones coming from exact diagonalization [Ref. 35], which exhibits that the ground state should be a quantum anomalous Hall (QAH) effect instead. These indeed come from small cluster calculations but combined with the arguments of Lieb-Schultz-Mattis theorem and its extension [Refs. 1 and 2] it shows that there should be a thermal transition at T_{th} to compatibilize their results, as they roughly propose in the text and more explicitly in the cartoon Figure 4. They cannot access such transition.

From what I could understand, no *direct* evidence in their results points to the connection between the QSH at $T > T_{th}$ and the QAH at $T = 0$ obtained on the same model using ED. The Hatsugai-Kohmoto model is used to provide support. Here the spin Chern number at quarter-filling gives $C_s = 1$, half of the non-interacting value at the same value of M at half-filling (another QSH). These calculations are obviously done at $T = 0$. However, they are already incompatible with the $T = 0$, ED results, which give a $C_s = 0$ (and Chern number $C = 1$). So one just hopes there should be a transition temperature to make these results (DQMC at large T and ED at $T = 0$) compatible.

Our response: We thank the referee for this comment. We address lower-temperature physics in two ways. First, we study the temperature evolution and U -dependence of the spin susceptibility for the flat-band Kane-Mele-Hofstadter-Hubbard (KM-HH) model using determinantal quantum Monte Carlo (DQMC) simulations. The results are shown in the updated Fig. 2 of the manuscript, also attached here in Fig. 1. The

extrapolation of it supports a finite-temperature phase transition. The divergence of the spin susceptibility indicates the spontaneous symmetry breaking towards Ising ferromagnetism and thus a quantum anomalous Hall (QAH) effect. Also, the spin susceptibility is enhanced by stronger interactions (Fig. 1b). Thus, the transition would also appear when U is sufficiently large to induce the high-temperature QSH features. Note QSH turns into the QAH effect in this transition as the temperature decreases. Also, note that both QSH and QAH have $C_s = 1$, though their Chern numbers are different ($C = 0$ for QSH and $C = \pm 1$ for QAH). QSH and QAH are closely related and it is spontaneous symmetry breaking that gives rise to this transition. These observations are from the DQMC simulations on the flat-band KM-Hubbard model. In the analytical calculation for the Hatsugai-Kohmoto (HK) interactions, the quarter-filling state was initially shown to have large degeneracy and spin Chern number $C_s = 1$ and Chern number $C = 0$. But later we found that the spin susceptibility diverges at $T = 0$, which means the ground state has an instability towards a ferromagnetic QAH state with spin Chern number $C_s = C_\uparrow - C_\downarrow = 1$ and Chern number $C = \pm 1$ ($C = 1$ for spin-up polarization and -1 for spin-down polarization).

Figure 1: Inverse spin susceptibility $1/\chi_s$ at quarter-filling ($\langle n \rangle = 1$) of the interacting flat-band generalized KM-HH model. Panel **a** contains the temperature evolution of $1/\chi_s$ at $U/t = 3$ with extrapolation to zero temperature. Panel **b** shows $1/\chi_s$ as a function of interaction strength at a fixed inverse temperature $\beta = 3/t$.

Figure 2: Estimated gap of the generalized KM-Hubbard model at quarter-filling as a function of the interaction strength U and the hopping phase ψ . These results are obtained from DCA simulations at a temperature of $\beta = 20/t$.

When the lower band becomes dispersive, we find that the gap doesn't necessarily open in the simulations for Hubbard interactions. To access the gap information, we resorted to the dynamical cluster approximation (DCA), a cluster version of the dynamical mean-field theory (DMFT). DCA has much milder finite-size effect and sign problems and thus it can access much lower temperatures. With DCA, we can explore how the dispersion of the band affects the gap opening at low temperatures. The detailed results and discussions are in the added "Gap opening" section of the manuscript and we summarize them below. The DCA results for the KM-HH model are attached in Fig. 2. As the band becomes more dispersive, namely $r = \Delta_0/W_{0-}$ decreases (where Δ_0 and W_{0-} are the non-interacting topological gap and lower bandwidth respectively), the critical U_c^{gap} to open a gap becomes larger and the asymptotic gap becomes smaller. In all cases, we set $\Delta_0 = 2$. When $r \leq 1$, we barely see a gap opening for $U \leq 12$. In the BHZ-Hubbard model, $r \leq 1$ in the topologically relevant parameter regime. As a consequence, we do not observe any obvious evidence for gap opening for $U < 12$. But we already observe the non-trivial topology emerging at quarter-filling at $U = 6$ and 8 (supplemental Fig. S8). Thus we conclude that our work points to a topologically non-trivial semimetal instead of an insulator at least for $U \leq 8$. Accompanied by the negative evidence for gap opening, we also observed that the temperature evolution of the spin susceptibility does not support a finite-temperature transition. This is in contrast to the ED study which observes the gap opening and ferromagnetic ordering for $U \gtrsim 4$ in the BHZ-Hubbard model. We conduct further DQMC simulations to show the finite-size effects can significantly bias the gap information for the cluster size in the ED study. A detailed discussion on this is given in the supplemental section "The analysis of finite size effect and gap opening for the BHZ-HH model with comparison to exact diagonalization".

The Reviewer wrote: Finally, if the authors are confident that their (unbiased) results unequivocally point out to a QSH effect at large T, what is the minimal temperature one needs to set in to see their appearance? In other words, the upper line in Fig. 4, separating a metal from the QSH, is a transition or a crossover? This is not a problem for DQMC, obviously.

Our response: We thank the referee for this question. The change from a metal to a high-temperature QSH feature is a crossover. The minimal temperature depends on U . For the flat-band KM-HH model shown in the updated Fig. 1 of the manuscript, $\beta = 3$ is sufficiently low to observe the QSH at $U = 12$, while at $U = 3$, $\beta = 4$ is needed to view the non-trivial topology. This is included in the supplemental section, "The crossover from metal to QAH/QSH features at high temperatures". In the updated manuscript, we dropped the phase diagram because it can not combine the emerging non-trivial topology, ferromagnetism, and gap opening in a simple way.

The Reviewer wrote: In summary, I don't think the authors are wrong, but there are some gaps in the logic to make their results compatible with other exact ones. I would be keen to hear from the authors before making an assessment about recommendation for publication.

Our response: We thank the referee for acknowledging our conclusion and providing suggestions for filling the logic gaps. As we answered above, the logical gaps have been filled with further DQMC and new DCA simulations.

The Reviewer wrote: Some points:

The $U = 0$ calculation of the BHZ-HH-(TRI) model can be done at any temperature in the DQMC (one field configuration suffices to obtain the physical properties). Have the authors tested that in the large β limit, one recovers the known results for this model? In particular, the slopes of the compressibility to give the spin Chern number? I suspect the features in Fig.1(a-d) to be sharper. I understand the motivation in the main text of using the same T as for the finite U calculations in Fig.1(e-h), but including it in the SM might be interesting.

Our response: We thank the referee for this suggestion. We tested that for $\beta = 20$, the calculation of the non-interacting BHZ-HH-TRI model (now we refer to it as the TRI compressibility of the BHZ-HH model for simplicity) recovers the known results. We include this benchmark in the updated supplemental section "Non-interacting results for the KM-HH model at lower temperatures".

The Reviewer wrote: The $T = 0$ phase at quarter filling should harbor a ferromagnetic ground-state. Large spin fluctuations are seen at this density for the $T > 0$ DQMC calculation. Owing to the Mermin-Wagner theorem, do they expect that there could be instead a quasi-long-range order at a finite temperature a la Kosterlitz-Thouless that defines T_{th} ? Or is this independent?

Our response: We thank the referee for this comment. As shown in Fig. 1, the extrapolation indicates a finite-temperature transition for the flat-band KM-Hubbard model. The Mermin-Wagner theorem says that continuous symmetries cannot be spontaneously broken at finite temperature in systems with sufficiently short-range interactions in dimensions $d \leq 2$. Here due to the spin-orbit coupling, the $SU(2)$ symmetry is lost and the model only conserves \hat{S}^z , namely the discrete Ising symmetry. Thus the spontaneous breaking of this symmetry at finite temperatures is allowed by the Mermin-Wagner theorem. A discussion on this is added to the updated manuscript.

The Reviewer wrote: When the authors study the Kane-Mele Hubbard model, they show in Fig. S1(b) a mapping of the sign vs. the density. From what I understand from Refs. PHYSICAL REVIEW B 84, 205121 (2011) and PHYSICAL REVIEW B 85, 115132 (2012), at half-filling in such a time-reversal model, one can show that the sign problem is absent. However, this figure shows that although is peaked at this density, it does not reach 1. What gives?

Our response: We thank the referee for pointing this out. Indeed, for the KM-Hubbard model, there is no sign problem exactly at half-filling. In that figure, we did not include exactly $\langle n \rangle = 1$, but only $\langle n \rangle = 0.99, 1.01$. After including exactly the data for $\langle n \rangle = 1$, its average sign indeed reaches 1, agreeing with the earlier quantum Monte Carlo (QMC) results. The updated figure for the KM-Hubbard model is shown in supplemental Fig. S1. However, after including the data at exactly $\langle n \rangle = 1$ in the BHZ-Hubbard model, we found that the average sign doesn't reach 1 at half-filling. Previous QMC-related studies (Phys. Rev. B 85 125113 and Phys. Rev. B 87 235104) only used DMFT on the BHZ-Hubbard model and found it sign-problem-free at half-filling. Then we further checked the average sign in DCA. We found that in DCA simulations, the average sign is exactly 1 for $N = 1$, namely the DMFT limit, in complete agreement with previous studies. Nonetheless the average sign decreases as the cluster size grows. A detailed report and explanation on this is in the supplemental section "Sign problem of DQMC and DCA simulations for the BHZ-HH model". A short answer is that the half-filled BHZ-Hubbard Hamiltonian is not invariant under a particle-hole transformation, while the half-filled KM-Hubbard model is effectively unchanged under the same transformation.

The Reviewer wrote: Minor points: Reproducibility is facilitated by knowing the details of the simulations. That is, what is the imaginary-time discretization used? The type of Hubbard-Stratonovich transformation? The typical sampling? Sentences such as "The results are striking." on page 6 bear no place in scientific texts. The community comes up with those if appropriate. Scultz -i Schultz in the abstract. Caption of Fig. 4 needs to explicitly state this is a cartoon or schematic representation of the expected physics, since none of it is based on actual data.

Our response: We thank the referee for this comment. We included the details of the simulations in the supplement and fixed the other points including minimizing the editorial remarks. As answered above, we dropped the phase diagram due to the lack of transparency.

RESPONSE TO REVIEWER #2

The Reviewer wrote: Motivated by recent experiments moire TMD bilayers, the authors study a model with a QSH ground state at half-filling that leads to a QAH state at quarter filling. The QSH state is smoothly connected to a non-interacting state and provided that the interaction does not exceed the gap remains robust.

Our response: In our work, there is no restriction on the interaction strength relative to the topological gap in the non-interacting system as in some papers in the literature: Phys. Rev. Lett. 108, 046806. Explicitly in this paper, the authors restrict themselves to the regime “ In all that follows, we assume that U is much smaller than the gap Δ_0 induced by a strong intrinsic spin-orbit coupling between the two pairs of bands.” Hence, explicitly the assumption is that the states that appear are adiabatically connected to the non-interacting system. This is not the case in any of our work. In fact, our observations are valid only when the interactions are the dominant scale and hence none of our results are adiabatically connected to the non-interacting limit. This is stated explicitly in the text.

The Reviewer wrote: The main surprise, according to the authors, is the emergence of a ferromagnetic quarter-filling QAH state. The work is interesting and the numerical analysis is thorough. However, I disagree with the framing of the results as surprising or qualitatively novel. The emergence of QAH states in models of narrow topological bands has been extensively studied over the past few years in the context of graphene moire systems and the main physics is reasonably well-understood as detailed below. Without placing the results in the context of recent findings in the field, it is difficult to assess its novelty. Thus, I do not recommend the manuscript for publication in its current form but am open to reconsider following suitable revisions. Let me elaborate below my concerns.

Our response: While the referee is correct that ferromagnetism has been found in topological models previously in the reference list provided, none of these papers studied the BHZ nor the KM model in the presence of interactions. As a result, they have not observed a 1/4-filled QAH or QSH when the interactions dominate in the KM-Hubbard and BHZ-Hubbard models. The numerical studies cited below on twisted-bilayer-graphene-related models based on the BM model with coulomb interactions and are focused either on charge neutrality (the QMC and DMRG study) or the true flat-band limit in which the interactions are much smaller than the gap from the remote dispersive bands.

The Reviewer wrote: 1. The model discussed by the authors for QSH (the BHZ model or even the Kane-Mele model) describes a 2D TI if only spinful time-reversal symmetry is preserved and a QSH insulator if $U(1)$ Sz symmetry is preserved. In limit where the interaction is smaller than the gap to the remote bands, the model describes a pair of opposite Chern bands labelled by opposite Sz which map to each other under time-reversal and are subject to strong interactions. This is identical to the problem considered in the context of graphene moire systems where $U(1)$ valley plays the role of Sz (see for example Bultinck et. al. PRL 124, 166601 (2020), Zhang et. al. PRB 99, 075127 (2019)).

Our response: We thank the referee for pointing out this limit. A key difference between the BM and KM/BHZ models is that while there are two gap scales in the BM model, the one between the flat Chern and dispersive bands, Δ_1 , and the other one between the $C = \pm 1$ Chern bands set by the sub-lattice potential, there is only one in the KM/BHZ models which is set by the spin-orbit. The sub-lattice potential gap in the BM model and the topological gap in the BHZ/KM models are related. This gap which we call Δ_0 sets a natural scale against which the interactions can be compared. Let us call the interaction scale, U . The referee is considering the case where either $U \ll \Delta_0$ or $U \ll \Delta_1$. While this is an interesting regime and covered previously in the literature, we NEVER work in either of these limits. In fact, the 1/4-filled topological

state never arises in this perturbative regime. Another difference between the BM and KM/BHZ models is that all the Chern bands in the BM model are flat whereas only the lower bands in KM can be made flat. Contrastly, both bands in BHZ are dispersive. The limit in which the lower bands of the KM model are flat and upper ones are dispersive is an explicit limit that we consider in our modeling of the experiments on TMD materials. Our physics arises entirely when the U -scale dominates over the bandwidth and the topological gap. The novelty of our work is that we find that the topology which heretofore existed at $1/2$ filling now shifts to $1/4$ filling when the interactions dominate. This is the novelty of this work.

The Reviewer wrote:

This problem is well-understood as a generalization of quantum Hall ferromagnetism where exact results can be obtained in the flat band limit (Bultinck et. al. PRX 10, 031034 (2020), Lian et. al. 103, 205414 (2021)). This picture has by now been verified by exact diagonalization (Xie et. al. PRB 103, 205416 (2021)), DMRG (Soejima et. al. 102, 205111 PRB (2020)), and QMC (Hofmann et. al. PRX 12, 011061 (2022)). The authors need to comment on the similarity/difference between the two problems and whether there is a reason to expect any qualitatively new aspect in the model they considered compared to the ones studied in graphene systems?

Our response:

As remarked earlier, we never work in the limit where $U \ll \Delta_0$. This said, we first address the treatment of the valley degree of freedom. While the valley degree of freedom can be viewed as a spin in a non-interacting system, such is not the case for an interacting system. In spinless models, treating the valley degree of freedom as an effective spin requires a long-range Coulomb interaction as the interaction must be written in momentum space. If spin is included explicitly, then Hubbard/HK interactions are the natural choice. Consequently, including Coulomb interactions in the BM model does not necessarily shed any light on the Hubbard/HK interactions in the KM/BHZ models. We expand on this below as we comment explicitly on the papers on the referee's list, all of which are now explicitly cited in the paper.

1.) Bultinck et. al. PRL 124, 166601 (2020) study the lowest-Landau level physics in the presence of interactions treated at the HF level and indeed they find ferromagnetism since such a symmetry-broken state is inevitable in Hartree-Fock (HF). A key in this work is the breaking of valley degeneracy induced by the alignment with h-BN. Consequently, the ferromagnetism and QAH in this context is quite distinct from the spontaneous symmetry breaking we have found here.

2.) Zhang et. al. PRB 99, 075127 (2019) is another HF study which cannot be used to address the general physics we find at $1/4$ -filling in the non-perturbative regime of the KM/BHZ-Hubbard models.

3.) Bultinck et. al. PRX 10, 031034 (2020) is another HF study.

4.) Lian et. al. 103, 205414 (2021) is another HF study.

5.) Xie et. al. PRB 103, 205416 (2021) performed an exemplary ED study on the flat-band BM model with Coulomb interactions. Here they found a series of $U(4)$ ferromagnets at $\nu = -1, -2, -3$ as well as Chern insulators all within an 8-band version of the BM model. As this model does not have spin-orbit coupling as in the BHZ/KM models in which only the S^z symmetry remains, the ferromagnet here is of the Ising type and hence distinct from the $U(4)$ ferromagnets in the BM study. In addition, the onset to ferromagnetism in the cases studied here at $1/4$ -filling does not always open a gap in contrast to the $U(4)$ ferromagnets of the interacting BM model. This is pointed out in the text.

6.) Soejima et. al. 102, 205111 PRB (2020) performed density-matrix-renormalization group studies and

found that the gapless state at half-filling in the spinless (and hence Mottless) Bisritzer-MacDonald model yields a quantum anomalous Hall state in the presence of Coulomb interactions. Since this model is spinless, there is not Mott physics present.

7.) QMC (Hofmann et. al. PRX 12, 011061 (2022)) performed beautiful Quantum Monte Carlo[46, 47] on the spinful model reveals a series of insulating states at half-filling (the CNP). This work does address the 1/4-filled state.

To make clear the difference between the flat-band and dispersive band results, we have now included a new section in the paper in which we study the evolution of the gap in the 1/4-filled state as a function of the interaction strength and dispersiveness of the lower band, as shown in Fig. 2. As is clear, in the dispersive case, we find non-trivial topology at 1/4-filling in which the gap does not open and no ferromagnetic ordering obtains. We explicitly lay out the criterion for the coincidence of both. In this case, the state is a semimetal. This is a substantially new result as it establishes that in the BHZ/KM-Hubbard models a regime exists in dispersive systems in which the topology is non-trivial but no gap exists (and hence no ferromagnetism) in the presence of interactions. There is no precursor to this result in the literature.

The Reviewer wrote: 2. In the discussion of the HK model, it is unclear what conclusions drawn by the authors are general and which ones are specific to the model.

Our response: This is an excellent question. In a series of papers (all cited) we have shown that because the HK model breaks the Z_2 symmetry of a Fermi liquid and since no local interactions destroy HK physics, the HK model represents a fixed point. In this context, not even Hubbard physics destroys HK. Hence, the model is completely general and as a result, the agreement with the Hubbard simulations is not an accident. For the logical organization of the paper, we put the simulations on the Hubbard model first in which the 1/4-filled state is shown to be special when the interactions dominate and then the general way of understanding why 1/4 is special is obtained from the HK model.

The Reviewer wrote: For instance, the authors mention there is a very large degeneracy (2^N) for the ground state manifold whereas the general expectation for a ferromagnet is $N+1$ for $SU(2)$ spin symmetry and 2 for $U(1)$ spin symmetry. Is this a consequence of the specific form of the interaction (HK) that is used?

Our Response:

Indeed the simplest HK model does have an extensive degeneracy which is lifted by an infinitesimal Zeeman field. This instability indicates ferromagnetism in the ground state and hence QAH. A different mechanism to lift the degeneracy was recently shown by B. Bradlyn's group (arXiv:2306.00221) simply by a 2-band version of the HK model. Here the degeneracy is now 2 rather than thermodynamic. The point of Bradlyn's paper is that since no real features of the model change in the multi-band limit, the degeneracy in the simplest HK model is really an artifact.

We hope this comprehensive response is satisfactory.

REVIEWERS' COMMENTS

Reviewer #1 (Remarks to the Author):

I commend the authors for a much clearer manuscript and for the detailed responses to my previous questions, either with new data included in the main text or in the revamped SM. I could also see an extensive response to the second referee. In the main text, one thing that becomes apparent is that the authors replace the orders in which the models are introduced, likely because, as they state,

"The tunability of bandwidths in the KM model (unlike the bands in the BHZ model which are always dispersive $W_0^+ = W_0^- \geq \Delta_0$) makes the KM model ideal for studying both flat-band and dispersive physics." Further, the new blue text on pages 2, 3 and 4 are now more instructive on the physics at play. New Fig. 2, on the spin susceptibility, is also very instructive.

I think that given the amount of investigation forming a logical body, there is enough evidence of the QSH physics at the 1/4 of the Kane-Mele and the Bernevig-Hughes-

Zhang model, which upon the spontaneous symmetry breaking of S_z symmetry gives way to a finite-T transition to a QAH phase. New Fig. 2 is evident in this regard. I'm positive other studies will follow, clarifying the results for an extensive set of parameters, etc., but this is fairly complete on the premise, and in the interest it brings to the community of quantum many-body systems encompassing topological characteristics. For this reason, I recommend the current version of the manuscript to be published in Nature Communications.

Minor comments:

- The sentence: "For $\langle n \rangle > 2$, the physics is essentially non-interacting as $U < W_0^+$ " can perhaps make non-interacting  weakly interacting.
- Caption of Figure 1 should describe the dashed green lines in panels (f) and (i). Same for Fig. 3(d).
- In the Sign problem section of the modified SM, the authors state that: "That being said, this problem may be studied using the sign-problem-free quantum Monte Carlo method in Majorana representation[10]". Could you do this with a spinful Hamiltonian? That is different from what was done in Ref. 10 there.

Reviewer #2 (Remarks to the Author):

The manuscript was improved by the revisions which included clarifying the parameter regime considered by the authors as well as comparison to related works in the context of moire systems. However, the authors mischaracterized some of the references I mentioned in my previous response (Bultinck et. al. PRX and Lian et. al. PRB) as purely HF studies. Bultinck et. al. contains both HF (Sec. III) and a strong coupling analytical argument (Sec. V) applicable in the limit of small bandwidth. Bian et. al. is purely analytical containing a similar strong coupling argument. Overall, as pointed out by the first referee, some details in the logic were missing in the original version which were mostly addressed in the current version. I recommend for publication after the addressing the comment above regarding the mischaracterization of some of the references.

RESPONSE TO REVIEWER #1

The Reviewer wrote: I commend the authors for a much clearer manuscript and for the detailed responses to my previous questions, either with new data included in the main text or in the revamped SM. I could also see an extensive response to the second referee. In the main text, one thing that becomes apparent is that the authors replace the orders in which the models are introduced, likely because, as they state, "The tunability of bandwidths in the KM model (unlike the bands in the BHZ model which are always dispersive $W_{0+} = W_{0-} \geq \Delta_0$) makes the KM model ideal for studying both flat-band and dispersive physics." Further, the new blue text on pages 2, 3 and 4 are now more instructive on the physics at play. New Fig. 2, on the spin susceptibility, is also very instructive.

I think that given the amount of investigation forming a logical body, there is enough evidence of the QSH physics at the 1/4 of the Kane-Mele and the Bernevig-Hughes-Zhang model, which upon the spontaneous symmetry breaking of S_z symmetry gives way to a finite-T transition to a QAH phase. New Fig. 2 is evident in this regard. I'm positive other studies will follow, clarifying the results for an extensive set of parameters, etc., but this is fairly complete on the premise, and in the interest it brings to the community of quantum many-body systems encompassing topological characteristics. For this reason, I recommend the current version of the manuscript to be published in Nature Communications.

Our response: We thank the referee for appreciating our revision and recommendation for publication.

The Reviewer wrote: Minor comments:

- The sentence: "For $\langle n \rangle > 2$, the physics is essentially non-interacting as $U < W_{0+}$ can perhaps make non-interacting \rightarrow weakly interacting.

Our response: We took the advice and made the change.

The Reviewer wrote: - Caption of Figure 1 should describe the dashed green lines in panels (f) and (i). Same for Fig. 3(d).

Our response: We took the advice and made the change.

The Reviewer wrote: - In the Sign problem section of the modified SM, the authors state that: "That being said, this problem may be studied using the sign-problem-free quantum Monte Carlo method in Majorana representation[10]". Could you do this with a spinful Hamiltonian? That is different from what was done in Ref. 10 there.

Our response: We thank the referee for this question. Ref.10 provides two spinful examples. One is the spinful single-band Hubbard model at half-filling, and the other is the spinful topological superconductor with Hubbard interaction. In principle, we think it may improve the sign problem of the systems studied here. We currently don't have the quantum Monte Carlo algorithm for the Majorana representation. But this is certainly an interesting project for future study.

RESPONSE TO REVIEWER #2

The Reviewer wrote: The manuscript was improved by the revisions which included clarifying the parameter regime considered by the authors as well as comparison to related works in the context of moire systems. However, the authors mischaracterized some of the references I mentioned in my previous response (Bultinck et. al. PRX and Lian et. al. PRB) as purely HF studies. Bultinck et. al. contains both HF (Sec. III) and a strong coupling analytical argument (Sec. V) applicable in the limit of small bandwidth. Bian et. al. is purely analytical containing a similar strong coupling argument. Overall, as pointed out by the first referee, some details in the logic were missing in the original version which were mostly addressed in the current version. I recommend for publication after the addressing the comment above regarding the mischaracterization of some of the references.

Our response: We thank the referee for pointing this out and have made the corrections in the manuscript.